# Phenolic Content and Bioactivity as Geographical Classifiers of Propolis from Stingless Bees in Southeastern Mexico

**DOI:** 10.3390/foods12071434

**Published:** 2023-03-28

**Authors:** Jorge Carlos Ruiz Ruiz, Neith Aracely Pacheco López, Estephania Guadalupe Rejón Méndez, Felipe Antonio Samos López, Luis Medina Medina, José Javier G. Quezada-Euán

**Affiliations:** 1Escuela de Nutrición, División de Ciencias de la Salud, Universidad Anáhuac-Mayab, Km 15.5 Carretera Mérida a Progreso, Int. Km 2 Carretera a Chablekal, Mérida 97310, Yucatán, Mexico; 2Centro de Investigación y Asistencia en Tecnología y Diseño del Estado de Jalisco, A.C., Subsede Sureste, Parque Científico Tecnológico de Yucatán, Km 5.5 Sierra Papacal-Chuburná Puerto, Mérida 97302, Yucatán, Mexico; npacheco@ciatej.mx; 3Departamento de Apicultura, Campus de Ciencias Biológicas y Agropecuarias, Universidad Autónoma de Yucatán, Apartado Postal 4-116, Mérida 97100, Yucatán, Mexico; esrejon_05@outlook.com (E.G.R.M.); felipe.samos18@outlook.com (F.A.S.L.); mmedina@correo.uady.mx (L.M.M.); javier.quezada@correo.uady.mx (J.J.G.Q.-E.)

**Keywords:** geographical, classification, stingless bees, propolis, phytochemicals, in vitro bioactivity potential

## Abstract

Propolis collected by stingless bees is a valuable biocultural resource and a source of bioactive compounds. Methodologies to establish both the geographic origin and the potential pharmacological activity of propolis of stingless bees are required to regulate their sustainable use. The aim of this study was to classify *Melipona beecheii* propolis according to its phenolic compound content and potential pharmacological activity, using in vitro assays and statistical methodologies of multivariate analysis, hierarchical cluster analysis, and principal component analysis. Propolis samples were collected from seven states in southeastern Mexico. Total phenolic content and flavonoids were determined spectrophotometrically, and antioxidant, anti-inflammatory, and antimicrobial activities were evaluated. Both total phenolic content and flavonoids, and in vitro bioactivity potential of propolis extracts showed significant variations. Multivariate analysis, hierarchical cluster analysis, and principal component analysis enabled us to distinguish and classify propolis produced by *M. beecheii* according to similarity in terms of total phenolic content, in vitro bioactivity potential, and geographical origin. This strategy could be used to establish regulations for sustainable use, marketing, and industrial applications.

## 1. Introduction

Stingless bees (Hymenoptera, Apidae: Meliponini) are the largest and the most diverse corbiculae eusocial bees [1,2]. Among the communities belonging to the Mayan ethnic group of the Yucatan Peninsula, meliponiculture is part of a strategy of multiple use of natural resources and constitutes an important activity in the Mayan peasant strategy due to its value of use-consumption and trade [3,4,5,6]. Propolis is a material produced by the corbiculate bees of the Apini and Meliponini tribes, which consists of resinous plant exudates that the bees collect from some plants and mix with waxes produced in specialized glands. Propolis produced by stingless bees is also called geopropolis, as some species mix the collected resins with materials such as soil and plant debris [7]. Propolis produced by stingless bees has long since been utilized in traditional medicine in Mexico, Brazil, Argentina, India, and Vietnam for improving health and for the treatment of wounds, burns, and skin conditions, as well as respiratory diseases [4,8,9]. Studies have reported that propolis exhibits various biological activities such as antioxidant, anti-inflammatory, antimicrobial, and anticancer, which are correlated with their pharmacological activities and uses [6,10,11].

Currently, because of their properties, propolis has diverse uses, for example, in wound treatment the antioxidants in propolis extracts can break the chain of free radicals that cause a detrimental effect to the wounded area. Furthermore, the antimicrobial properties of propolis can overcome bacterial contamination and thus improve the healing rate. Moreover, the anti-inflammatory attribute in propolis extracts can protect the tissue from highly toxic inflammatory mediators [12,13,14,15]. On the other hand, recent studies with stingless bee products have shown a higher rate of epithelialization in wounds and greater anti-inflammatory and antimicrobial effects than those of European or honeybees [16]. Hydroalcoholic propolis extracts administered at 50 and 200 mg/kg to mice, reversed the pattern of inflammatory cells in the lung and decreased the influx of polymorphonuclear inflammatory cells to parenchyma [17]. On the other hand, ethanolic extracts (1000 mg/kg) of propolis from *Scaptotrigona jujuyensis* Schrottky and *Tetragonisca fiebrigi* Schwarz, significantly reduced the carrageenan-induced edema and cotton pellet-induced granuloma formation 3 h post-dosing in Wistar male rats. In the same study of ammonia liquor-induced cough, both propolis extracts significantly enhanced the latent period and reduced cough frequency [18].

There are several reports on the characterization of the phenolic phytochemical content and the biological activities of propolis produced by honeybees and stingless bees [19,20,21,22]. However, there is a need for rapid, reliable, and low-cost methodologies that allow determination of the traceability and authenticity of propolis collected by stingless bees. Determination of physicochemical properties entails limitations due to the high variability between product samples and the need for a large amount of sample for the evaluation of various parameters, which translates into high time and cost consumption [23,24]. Considering the potential therapeutic applications of propolis, the methodologies applied to its study should focus on bioactivity and phytochemical compounds associated with bioactivity [25]. Otherwise, chemometrics consists of a multivariate data analysis method that has been used for the analysis of fingerprints and chemical profile of biological samples [26,27]. This study aimed to evaluate the use of phenolic and flavonoid contents, and in vitro assays of antioxidant, anti-inflammatory, and antibacterial activities in combination with a chemometric approach for the classification of *Melipona beecheii* propolis from southeastern Mexico. Differentiation of propolis produced by *Melipona beecheii* could help identify its geographical origin, guarantee its quality, and authenticity for potential therapeutic applications.

## 2. Materials and Methods

### 2.1. Reagents and Materials

All reagents used were of analytical grade. All dilutions were made using distilled water. All glassware and plastic bottles used were previously decontaminated by immersion in a 10% HNO_3_ solution for 24 h and washed with deionized water prior to use.

### 2.2. Propolis

Propolis samples were collected from a total of 35 *Melipona beecheii* hives from 12 locations in 6 southeastern Mexican states (Table 1), including Campeche (9 samples), Oaxaca (1 sample), Quintana Roo (9 samples), Tabasco (6 samples), Veracruz (3 samples), and Yucatán (7 samples). Propolis was collected from domesticated hives from January to August 2019. Samples were stored in amber bottles at −4.0 °C prior to analysis.

### 2.3. Propolis Extraction

Propolis samples were ground in a marble mortar and 1 g of pulverized propolis was weighed and 30 mL of ethanol (70%, *v*/*v*) added. The mixture was kept under mechanical agitation at room temperature and in the absence of light for 24 h. Then, the mixture was filtered (Whatman filter paper No. 4), and the solid was re-extracted under the same conditions as reported. After the second extraction, the extracts were combined in a 50-mL volumetric flask and the volume was adjusted with ethanol (70%, *v*/*v*). The extraction was stored in amber glass bottles and kept frozen until further use.

### 2.4. Total Phenolic and Flavonoids Quantification

Total phenolic content was determined in the ethanolic extracts by the method based on the reaction of phenolics with Folin–Ciocalteu reagent. Absorbance was determined at 760 nm, using gallic acid as a standard for the calibration curve. Results were expressed as mg equivalents of gallic acid (GA)/g of propolis. For the determination of flavonoids, these compounds reacted with AlCl_3_ in alkaline medium. Absorbance was determined at 510 nm, using catechin as standard for the calibration curve. Results were expressed as mg equivalents of catechin (C)/g of propolis [27].

### 2.5. Antioxidant Assays

#### 2.5.1. DPPH (2,2-Diphenyl-1-picrylhydrazyl) Free Radical Scavenging

A volume of 500 μL of propolis extract was mixed with 500 μL of DPPH (0.1 mM, 95% ethanol). Absorbance was determined at 517 nm. The percentage of the DPPH free radical scavenging was calculated using Equation (1):(1)$$\mathrm{DPPH}\;\mathrm{scavenging}\;\mathrm{effect}\;\left(\%\right)=\frac{A0-A1}{A0}\times 100$$
where A0 = absorbance of blank and A1 = absorbance in presence of propolis extracts. Ascorbic acid was used as control [28].

#### 2.5.2. ABTS (2,2-Azinobis-(3-ethylbenzothiazoline-6-sulfonate)) Free Radical (Hydrophilic) Scavenging

The antioxidant activity was performed through the ABTS method where the ABTS^•+^ radical was formed by the reaction of 7 mmol/L of ABTS with potassium persulfate (140 mmol/L), incubated at 25 °C in the dark for 16 h. The radical was diluted with phosphate-buffered saline and resulted in an absorbance of 0.700 ± 0.200 at 734 nm. Under dark conditions, a volume of 3.0 mL of the ABTS^•+^ solution was added to 30 µL of propolis extract and the absorbance was read at 734 nm with a spectrophotometer after 6 min. The percentage of the ABTS free radical scavenging was calculated using Equation (2):(2)$$\;\mathrm{ABTS}\;\mathrm{scavenging}\;\mathrm{effect}\;\left(\%\right)=\frac{A0-A1}{A0}\times 100$$
where A0 = absorbance of blank and A1 = absorbance in presence of propolis extracts. Ascorbic acid was used as control [28].

#### 2.5.3. Metal Chelating Ability

Chelating activity was determined using the pyrocatechol violet reagent. Briefly, 1.0 mL of sodium acetate buffer (100 mM, pH 4.9), 100 μL of Cu (II) standard solution (1.0 mg/mL), and 100 μL of propolis extract were homogenized in a microtube. The mixture reacted during 5 min at room temperature and 25 μL of a pyrocatechol violet solution (4.0 mmol/L) was then added. Absorbance was determined at 632 nm. Chelating activity was calculated using Equation (3):(3)$$\mathrm{Chelating}\;\mathrm{activity}\;\left(\%\right)=1-\frac{\mathrm{SA}}{\mathrm{BA}}\times 100$$
where SA = sample absorbance and BA = blank absorbance. Ascorbic acid was used as control [29].

#### 2.5.4. Ferric Reducing Antioxidant Power

This method is based on the reduction of potassium ferricyanide (Fe^3+^) in the presence of an antioxidant (Fe^2+^) to form the blue complex K[FeII(CN_6_)], which absorbs at 700 nm. First, 200 μL of propolis extract, 500 μL of phosphate buffer (0.2 M, pH 6.6), and 500 μL of potassium ferricyanide (1%) were homogenized in a test tube. The test tube was then incubated at 50 °C for 20 min. Subsequently, 500 μL of trichloroacetic acid 10% (*w*/*v*) was added, and the tube was centrifuged at 3000× *g* for 10 min. An aliquot of 500 μL of the supernatant was dissolved in an equal amount of distilled water and immediately 500 μL of ferric chloride (0.1%) was added. Absorbance was determined at 700 nm. The percentage inhibition of the K[FeII(CN_6_)] complex formation was calculated using Equation (4):(4)$$\mathrm{Ferric}\;\mathrm{reducing}\;\mathrm{antioxidant}\;\mathrm{power}\;\left(\%\right)=\frac{A0-A1}{A0}\times 100$$
where A0 was the absorbance of the control, and A1 of the mixture containing propolis extract. Ascorbic acid was used as control [28].

### 2.6. Anti-Inflammatory Assays

#### 2.6.1. Inhibition of Protein Thermal Denaturation

The reaction mixture consisted of 500 μL of propolis extract and 500 μL of 5% (*w*/*v*) albumin solution. The mixture was incubated at 37 °C for 20 min, and then the temperature was increased to 70 °C for 5 min. Turbidity was determined at 660 nm. The percentage inhibition was calculated using Equation (5):(5)$$\mathrm{Inhibition}\;\mathrm{of}\;\mathrm{protein}\;\mathrm{thermal}\;\mathrm{denaturation}\;\left(\%\right)=\frac{\mathrm{CA}-\mathrm{SA}}{\mathrm{CA}}\times 100$$
where: CA = control absorbance and SA = sample absorbance. Acetylsalicylic acid was used as control [30].

#### 2.6.2. Cell Membrane Stabilization

Fresh human blood samples were obtained from healthy volunteers, the samples were available in accordance with the provisions of Mexican regulations [31]. From the fresh blood sample, 1.0 mL was taken, and 1.0 mL of saline solution (0.9%) added; this mixture was centrifuged at 3000 rpm for 10 min. The cell pack (leukocytes, platelets, and erythrocytes) was washed again with saline solution (0.9%); this procedure was performed for a total of 5 times, taking the cell pack at the end. At the end of washings, the amount of precipitate remaining was reconstituted in a 1:1 ratio with a saline solution (0.9%). From the 1% (*v*/*v*) red blood cell solution, 500 μL was taken and 500 μL of the sample added, this reaction was incubated at 56 ◦C for 30 min, and centrifuged at 2500 rpm for 5 min. The absorbance of the supernatant was measured at 560 nm. The stabilization of the red blood cell membrane was calculated using Equation (6):(6)$$\mathrm{Cell}\;\mathrm{membrane}\;\mathrm{stabilization}\;\left(\%\right)=1-\frac{\mathrm{CA}-\mathrm{SA}}{\mathrm{CA}}\times 100$$
where: CA = control absorbance and SA = sample absorbance. Acetylsalicylic acid was used as control [32,33].

#### 2.6.3. Hemolysis Assay

Whole fresh human blood from healthy volunteers (15 mL) was collected in 2,2′,2″,2‴-(Ethane-1,2-diyldinitrilo) tetraacetic acid (EDTA) tubes and centrifuged for 10 min at 1000× *g* at 4 °C. The plasma was removed, and the obtained red blood cells (RBCs) were suspended in 10 mM Phosphate-buffered saline (PBS) (pH 7.4). The erythrocytes were washed 3 times with PBS and re-suspended in PBS to obtain a solution at 4%. 1 mL of this suspension was mixed with different concentrations of propolis extracts and added to 7.5 mM of H_2_O_2_ prepared in PBS. After incubation for 120 min at 37 °C, the mixture was centrifuged at 1000× *g* for 5 min. Finally, the absorbance of the supernatant was determined at 540 nm. Ascorbic acid was used as control. Hemolysis was calculated using Equation (7):(7)$$\mathrm{Hemolysis}\;\mathrm{inhibition}\;\left(\%\right)=\frac{A2-A1}{A2-A0}\times 100$$
where, A0 = absorbance of RBC suspension in PBS, A1 = absorbance of samples with RBC suspension in PBS/H_2_O_2_, A2 absorbance of RBC suspension in PBS and H_2_O_2_. Acetylsalicylic acid was used as control [34].

### 2.7. Antibacterial Assay

#### Evaluation of Minimum Inhibitory Concentrations (MIC)

The antibacterial activity of the propolis extracts was determined in the following way: Bacterial colonies from the agar medium were dissolved in test tubes with normal saline solution to obtain inoculum suspensions with a concentration of 1 × 10^6^ bacteria. The inoculum suspensions were distributed in a 96-well microtiter plate containing a two-fold serial dilution of the propolis samples. The MIC value was determined as the lowest concentration of propolis extract that inhibited bacterial growth after incubation at the optimal temperature. Streptomycin or vancomycin were used as positive controls [35,36].

### 2.8. Statistical Analysis

Correlation analysis via the bivariate technique was used to calculate the relationship between various parameters. Results were expressed as Pearson correlation coefficients (r). Multivariate analyses using hierarchical cluster analysis and principal component analysis were performed to classify propolis samples based on phytochemical contents and biological properties using XLSTAT Premium for Excel statistical software (Lumivero Denver, CO, USA). Redundant and/or fewer discriminating variables were removed as they could affect the predictive ability of chemometrics. The selected variables were proceeded by hierarchical cluster analysis (HCA) and principal component analysis (PCA) analyses.

## 3. Results

### 3.1. Total Phenolic and Flavonoids Quantification

Total phenolic contents ranged from 18.65 to 226.14 μg equivalents of gallic acid/mL extract (Table 2). The average content of total phenols varied significantly (*p* < 0.05) between states, with the state of Yucatan exhibiting the highest values followed by the states of Veracruz, Campeche, Tabasco, Oaxaca, and Quintana Roo.

Variations in the content of phenols occur even between locations in the same state, for example the average content of total phenols of propolis from Espita (Yucatan) is 3.8 times higher than that of samples from Mama (Yucatan). Propolis samples from other states included in the sampling show the same behavior. With respect to the flavonoid content, these ranged from 6.51 to 59.85 μg equivalents of catechin/mL extract (Table 2). The average content of flavonoids varied significantly (*p* < 0.05) between states, with the state of Quintana Roo exhibiting the highest values followed by the states of Veracruz, Oaxaca, Tabasco, Campeche, and Yucatán. Variations in the content of flavonoids occur even between locations in the same state, for example the average content of flavonoids of the propolis from Espita (Yucatan) is two times higher than that of the samples from Mama (Yucatan). Propolis samples from the other states included in the sampling show the same behavior.

### 3.2. Antioxidant Assays

DPPH free radical scavenging activity for propolis extracts ranged from 8.01 to 95.94% (Table 2). The average percentage of radical scavenging (DPPH) activity varied significantly (*p* < 0.05) between states, with the state of Oaxaca exhibiting the highest values followed by the states of Yucatan, Tabasco, Veracruz, Campeche, and Quintana Roo. Otherwise, ABTS assay has been used to determine the antioxidant capacity of food products. ABTS free radical scavenging activity for propolis extracts ranged from 1.14 to 49.11% (Table 1). The average percentage of radical scavenging (ABTS) activity varied significantly (*p* < 0.05) between states, with the state of Yucatan exhibiting the highest values followed by the states of Campeche, Oaxaca, Tabasco, Quintana Roo, and Veracruz. Metal chelating ability for propolis extracts ranged from 45.92 to 93.15% (Table 1). The average percentage of metal chelating ability varied significantly (*p* < 0.05) between states, with the state of Oaxaca exhibiting the highest values followed by the states of Veracruz, Tabasco, Quintana Roo, Campeche, and Yucatán. Ferric reducing antioxidant power for propolis extracts ranged from 18.23 to 89.02% (Table 2). The average percentage of ferric reducing antioxidant power varied significantly (*p* < 0.05) between states, with the state of Tabasco exhibiting the highest values followed by the states of Oaxaca, Yucatan, Campeche, Veracruz, and Quintana Roo.

### 3.3. Anti-Inflammatory Assays

Cell membrane stabilization of propolis extracts ranged from 33.48 to 83.70% (Table 2). The average percentage of cell membrane stabilization varied significantly (*p* < 0.05) between states, with the state of Yucatan exhibiting the highest values followed by the states of Campeche, Quintana Roo, Tabasco, Veracruz, and Oaxaca. Second, hemolysis of propolis extracts ranged from 16.30 to 66.52%. The average percentage of hemolysis varied significantly (*p* < 0.05) between states, with the state of Yucatan exhibiting the lowest values followed by the states of Campeche, Quintana Roo, Tabasco, Veracruz, and Oaxaca. Results presented by various phenolic extracts in the inhibition of protein thermal denaturation assay have been correlated with the effect of anti-inflammatory drugs on pathologies such as rheumatoid arthritis, diabetes, and cancer [34]. Inhibition of protein thermal denaturation of propolis extracts ranged from 8.69 to 14.66% (Table 1). The average percentage of inhibition of protein thermal denaturation varied significantly (*p* < 0.05) between states, with the state of Veracruz exhibiting the highest values followed by the states of Tabasco, Campeche, Quintana Roo, Yucatan, and Oaxaca.

### 3.4. Antibacterial Assay

Various studies report that stingless bee propolis extracts exhibit antimicrobial activity against Gram-positive and Gram-negative pathogenic bacteria. However, they show greater effectiveness against Gram-positive bacteria. Some extracts have even been effective against strains of *S. aureus*, *E. faecalis*, *E. coli,* and *P. aeruginosa* resistant to antibiotics such as methicillin, vancomycin, cephalosporin, and imipenem [4,37,38]. In this study, the extracts of propolis from Espita (Yucatan), Zoh-Laguna (Campeche), and Felipe Carrillo Puerto (Quintana Roo) showed activity against Staphylococcus aureus and Salmonella typhi, samples from Espita, Mexico, Yucatan being the most active in inhibiting bacterial growth at a lower concentration (Table 1). In the case of extracts of propolis from Centauros del Norte (Campeche), they only showed activity against *Salmonella typhi*. Regarding antibacterial activity, the solvent used in the extractive process influences the type of compounds extracted, which affects the inhibition on bacterial growth. Extracts obtained with methanol, ethyl acetate, and hexane show different effects on *E. coli* and *S. aureus* in disk diffusion assays [39].

### 3.5. Chemometrical Analysis

#### 3.5.1. Pearson Correlation

Pearson correlation is a number between −1 and +1 and measures the degree of linear relationship between two parameters. A correlation of positive values indicates a positive (increasing) linear relationship, while a negative correlation indicates a negative (decreasing) linear relationship. Table 3 shows Pearson correlations between phenolic compound content, antioxidant, anti-inflammatory, and antibacterial properties of propolis samples.

Strong and significant positive correlations were found for total phenolic content with both DPPH free radical (hydrophobic) scavenging and ABTS free radical (hydrophilic) scavenging (*p <* 0.05). Metal chelating activity was negatively correlated with both DPPH free radical (hydrophobic) scavenging and ABTS free radical (hydrophilic) scavenging (*p <* 0.05). The same as for cell membrane stabilization and hemolysis (*p <* 0.05).

#### 3.5.2. Hierarchical Cluster Analysis

For the successive analyses of hierarchical cluster (HCA) and principal component (PCA), highly correlated and redundant variables could be removed by considering the F-ratio values [26,40]. Total phenolic content, flavonoid content, DPPH free radical scavenging, ABTS free radical scavenging, metal chelating activity, ferric reducing antioxidant power, cell membrane stabilization, and hemolysis were selected and retained for HCA and PCA whereas inhibition of protein thermal denaturation, Gram-positive (Staphylococcus aureus), and Gram-negative (*Salmonella typhi*) minimum inhibitory concentrations with low F-ratio values were reduced. Figure 1 shows the dendrogram obtained from HCA based on variables highly correlated considering the F-ratio values. Vertical lines in the dendrogram represent linkage distance between samples while horizontal lines represent propolis samples. Small linkage distance means high similar characteristics. Propolis samples from different origins were grouped together into their varieties and five clusters were obtained at linkage distance of 0.28 (Similarity). First cluster comprised propolis samples from EY; these samples come from areas where deciduous forest vegetation predominates. Additionally, Yucatan has a high endemism of plant species distributed in specific geographic areas [41]. This would explain why the samples from MY, even though they were from the same state, were classified in a different cluster (Figure 2). The second cluster comprised propolis samples from MY, CNC, ZLC, and TQR; samples from these localities come from three different states. However, they share the same type of predominant vegetation: evergreen forest and semi-deciduous forest. Third cluster comprised propolis samples from IKC, ZLC, SMT, and SJBO; samples from these localities come from three different states and are geographically far apart from each other. However, they are found in areas where the predominant vegetation has been modified for agricultural use, leaving some remnants of evergreen forest and semi-deciduous forest. The fourth cluster comprised propolis samples from RT; predominant vegetation where these samples come from is represented by cultivated areas, spiny forest, and evergreen forest. Finally, the fifth cluster comprised propolis samples from BQR, FCPQR, and NVV; BQR and FCPQR collected in Quintana Roo were obtained in semi-deciduous forest areas, while NVV was collected in an area of cultivated vegetation and semi-deciduous forest.

#### 3.5.3. Principal Component Analysis

The PCA was applied to classify all 35 propolis samples following its geographical origin using tested variables (Figure 2). The PC1 (50.53%) and PC2 (26.78%) which attributed for 77.31% of the total variance were extracted to determine the best variables that could classify the propolis samples based on their geographical origin. Total phenolic content (PC1) has a direct influence on the variables: upper cell membrane stabilization (right quadrant), DPPH and ABTS radical scavenging, and ferric reducing antioxidant power; all of these are in the lower right quadrant. Flavonoid content (PC2) has a direct influence on the variables: metal chelating activity and hemolysis in the lower left quadrant. The PCA biplot showed that some samples collected from MY, IKC, ZLC, and RT were situated in in the upper right quadrant (Figure 2). Some samples collected from MY, CNC, IKC, TQR, BQR, FCPQR, and RT were situated in the upper left quadrant. Samples collected from EY and some samples collected from ZLC were situated in the lower right quadrant. Finally, some samples collected from FCPQR, SMT, and NVV were situated in the lower left quadrant. Propolis samples located on the right side of the quadrant are defined by the variables influenced by the total content of phenols; such is the case of DPPH free radical scavenging, ABTS free radical scavenging, ferric reducing antioxidant power, and cell membrane stabilization. Moreover, propolis samples located on the left side of the quadrant are defined by the variables influenced by the flavonoid content; such is the case of metal chelating capacity and hemolysis.

## 4. Discussions

Composition of propolis is related to geographic location, predominant vegetation, season of the year, climatic conditions, and species of bee; it can be composed of balms (50%), waxes (30%), essential oils (10%), and pollen (5%) [13,42,43]. In this study, conventional extractions of phenolic-type compounds were carried out since these are easily quantifiable phytoconstituents through reproducible and low-cost analytical techniques, in addition to the fact that studies have reported their various bioactivities. The collected samples come from localities with different types of predominant vegetation such as cloud forest, temperate forest, humid forest, sub-humid forest, and xeric scrub. Our results indicated that content of total phenolic compounds and flavonoids could be related to the type of vegetation predominant in the locations where the propolis was collected. The results obtained in this study coincide with those reported by other authors for classification and authentication of propolis from stingless bees collected in tropical forests in Asia and Brazil [11,13,44,45,46,47]. The correlation analysis between the contents of phenols and flavonoids was significantly negative. A study conducted on propolis stingless bees from Malaysia indicated the opposite behavior [45]. In this sense, the differences observed in our study may be due to factors such as bee species, predominant vegetation, and extraction methods.

For biological activity evaluation, our results indicate that the propolis extracts with the highest total phenol content showed the highest values in the free radical scavenging and ferric reducing antioxidant power assays, while extracts with high flavonoid content showed highest values for chelating capacity assay. The total content of phenols and flavonoids has a decisive influence on the antioxidant capacity as reported by authors for studies carried out with propolis collected in Turkey and Brazil [46,47]. These results agree with the Pearson correlation analysis, which indicated a positive correlation for total phenolic content with free radical scavenging ferric reducing antioxidant power assays. On the other hand, flavonoids are negatively related to chelating capacity. Results of anti-inflammatory and antibacterial assays did not show a relationship with total phenolic content and flavonoids.

The PCA biplot indicated that the samples collected in EY and ZLC are grouped in the quadrant of antioxidant assays for free radical scavenging and reducing power; propolis samples from Espita (Yucatan) stand out since they were also grouped in an individual cluster by Agglomerative Hierarchical Cluster Analysis. These results indicate that it is feasible to classify propolis from different locations as reported in studies for propolis classification and authentication [48,49,50].

In our study, propolis extracts from all locations exhibited anti-inflammatory activity. According to that reported by some studies, the anti-inflammatory activity is related to the content of phenolic compounds and flavonoids [51,52]. However, in our study, the statistical analysis of the data obtained from the in vitro assays indicated that they are not significantly correlated with the contents of phenolic compounds and flavonoids in the extracts. Additionally, the PCA biplot indicated that the anti-inflammatory tests did not agglomerate propolis samples from any locality. Although the propolis extracts of all the samples exhibited anti-inflammatory properties in assays, it is necessary in the future to apply other methodologies that allow delving into the mechanisms involved in the inflammatory processes; the compounds involved in the biological activities must also be purified and identified by means of instrumental methods.

Results obtained in this study indicate that not all the biological activities evaluated by means of in vitro assays were significant in classifying propolis from different localities in southeastern Mexico. In this sense, the content of phenolic and flavonoid compounds, as well as the antioxidant properties are the most appropriate to carry out a classification in terms of geographical origin. The anti-inflammatory and antibacterial properties should be complemented with studies of phenolic profiles and other statistical methodologies to establish their usefulness as propolis classifiers.

## 5. Conclusions

The results obtained from the quantification of phenolic compounds and flavonoids indicate the influence that geographical origin has on the content of these secondary metabolites in propolis of *Mellipona beecheii*. Results of in vitro assays of antioxidant, anti-inflammatory, and antibacterial properties of propolis extracts indicate that the total content of phenolic compounds and flavonoids of propolis extracts are correlated with biological activity. The use of extensive sampling in conjunction with correlation analysis, hierarchical cluster analysis, and principal component analysis for pattern recognition of phenolic phytochemical content and biological properties showed that propolis samples produced by *Melipona beecheii* stingless bee could be classified according to their geographical origin. The results of this study could provide further guidance with respect to determining quality parameters for the control of stingless bee products, with the purpose of generating regulatory standards in the future that promote the sustainable use and marketing of this type of product.

## Figures and Tables

**Figure 1 foods-12-01434-f001:**
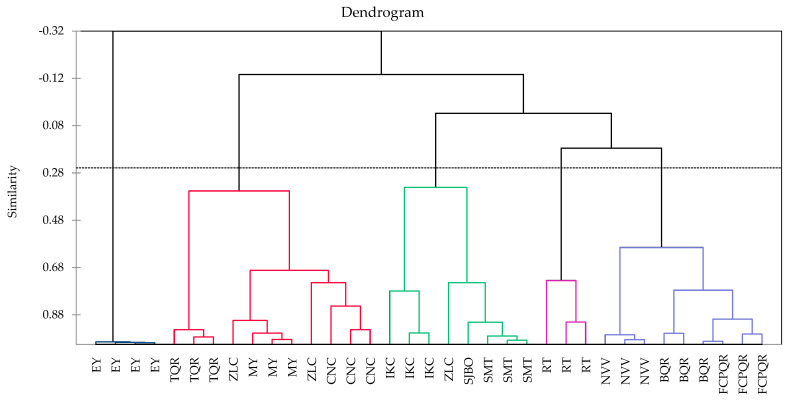
Dendrogram from Agglomerative Hierarchical Cluster Analysis of 35 propolis samples using phenolic and flavonoid contents, and in vitro assays of antioxidant, anti-inflammatory, and antibacterial properties. EY = Espita, Yucatan; MY = Mama, Yucatan; CNC = Centauros del Norte, Campeche; IKC = Ich-Ek, Campeche; ZLC = Zoh-Laguna, Campeche; BQR = Bacalar, Quintana Roo; FCPQR = Felipe Carrillo Puerto, Quintana Roo; TQR = Tihosuco, Quintana Roo; RT = Reforma, Tabasco; SMT = San Marcos, Tabasco; NVV = Ejido Nicolás Bravo, Veracruz; SJBO = San Juan Bautista, Oaxaca.

**Figure 2 foods-12-01434-f002:**
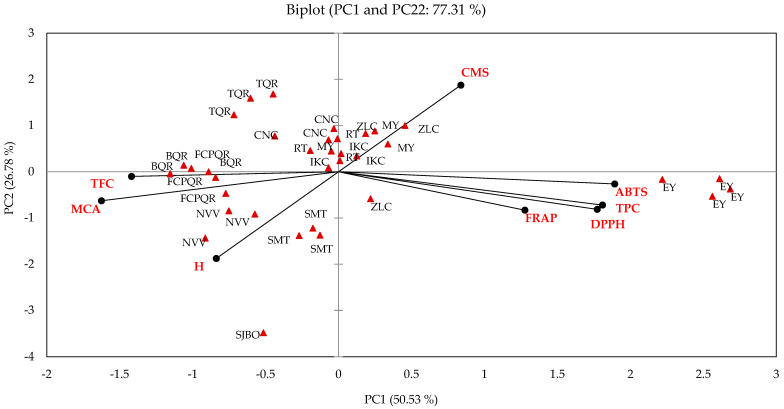
The score plot for PC2 versus PC1 from PCA based on content of phenolic compounds and in vitro assays of bioactivity properties for propolis classification following its geographic location. Italic codes indicate samples from different locations. Bold codes indicate determined variables. TPC = Total phenolic content. F = Flavonoids. DPPH = DPPH free radical (hydrophobic) scavenging, ABTS = ABTS free radical (hydrophilic) scavenging, MCA = Metal chelating activity, FRAP = Ferric reducing antioxidant power, CMS = Cell membrane stabilization, H = Hemolysis. EY = Espita, Yucatan; MY = Mama, Yucatan; CNC = Centauros del Norte, Campeche; IKC = Ich-Ek, Campeche; ZLC = Zoh-Laguna, Campeche; BQR = Bacalar, Quintana Roo; FCPQR = Felipe Carrillo Puerto, Quintana Roo; TQR = Tihosuco, Quintana Roo; RT = Reforma, Tabasco; SMT = San Marcos, Tabasco; NVV = Ejido Nicolás Bravo, Veracruz; SJBO = San Juan Bautista, Oaxaca.

**Table 1 foods-12-01434-t001:** The propolis samples, collected states, locations, symbols, and geographical coordinates.

State	Locations	Symbol	Geographical Coordinates
Yucatan	Espita	EY	21°00′46″ N, 88°18′17″ W
Yucatan	Mama	MY	20°28′38″ N, 89°21′54″ W
Campeche	Centaruros del Norte	CNC	18°12′16″ N, 91°32′9″ W
Campeche	Ich-Ek	IKC	19°44′0″ N, 89°58′1″ W
Campeche	Zoh-Laguna	ZLC	18°35′14″ N, 89°25′1″ W
Quintana Roo	Bacalar	BQR	18°40′42″ N, 88°23′33″ W
Quintana Roo	Felipe Carrillo Puerto	FCPQR	19°34′43″ N, 88°02′43″ W
Quintana Roo	Tihosuco	TQR	20°11′45″ N, 88°22′25″ W
Tabasco	Reforma	RT	17°52′00″ N, 93°14′00″ W
Tabasco	San Marcos	SMT	18°02′25″ N, 93°00′57″ W
Veracruz	Ejido Nicolas Bravo	NVV	18°40′24″ N, 97°24′54″ W
Oaxaca	San Juan Bautista	SJBO	16°30′35″ N, 90°20′50″ W

Samples from three domesticated hives of Melipona beecheii were collected in each locality.

**Table 2 foods-12-01434-t002:** Total phenolic content, flavonoid content, and in vitro assays of antioxidant, anti-inflammatory, and antibacterial activities of propolis extracts from Melipona beecheii hives by locality.

Sample	TPC	FC	DPPH	ABTS	MCA	FRAP	CMS	H	IPTD	GP	GN
MY	51.52	9.57	16.03	5.54	70.81	47.23	66.18	33.82	12.06	ND	ND
MY	56.15	8.18	20.51	9.12	74.86	57.34	79.52	20.48	11.57	ND	ND
MY	49.54	6.51	24.22	12.32	72.29	61.22	74.15	25.85	8.69	ND	ND
EY	196.03	9.78	92.91	44.88	47.38	84.93	71.70	28.30	8.87	0.39	0.39
EY	226.14	12.76	93.63	49.11	47.38	74.01	75.30	24.70	10.51	0.09	1.45
EY	207.11	8.04	95.94	41.14	45.92	76.02	78.33	21.67	11.47	ND	ND
EY	208.88	13.60	90.19	34.94	59.09	76.78	80.04	19.96	11.45	ND	ND
TQR	20.95	48.54	8.01	8.35	69.29	19.85	82.90	17.10	11.33	ND	ND
TQR	24.74	46.68	13.14	3.12	76.74	27.95	79.16	20.84	10.94	ND	ND
TQR	23.20	39.39	14.74	3.64	64.23	18.23	83.70	16.30	11.30	ND	ND
BQR	18.65	39.26	13.11	3.73	87.22	21.62	53.17	46.83	11.72	ND	ND
BQR	19.83	45.62	11.29	5.22	79.15	21.36	55.12	44.88	12.15	ND	ND
BQR	24.74	50.38	19.59	2.90	76.79	37.88	56.32	43.68	12.57	ND	ND
FCPQR	39.60	47.55	22.97	5.22	85.95	55.50	53.62	46.38	10.00	12.49	12.49
FCPQR	36.82	59.85	13.21	4.20	83.91	41.99	60.52	39.48	11.63	6.25	3.12
FCPQR	45.67	48.59	15.24	3.46	85.45	48.94	58.53	41.47	10.93	ND	ND
IKC	37.31	12.37	13.07	20.21	84.42	57.39	70.50	29.50	12.00	ND	ND
IKC	49.20	17.53	21.33	17.51	78.88	59.16	70.32	29.68	12.56	ND	ND
IKC	39.39	15.91	10.36	27.88	90.69	55.02	65.70	34.30	10.86	ND	ND
CNC	38.27	16.71	13.14	3.22	75.79	27.62	69.13	30.87	11.69	ND	6.25
CNC	35.08	18.44	13.18	12.46	63.21	33.90	71.51	28.49	12.54	ND	6.26
CNC	42.65	19.02	13.75	5.92	60.33	41.50	67.12	32.88	12.86	ND	ND
ZLC	61.94	10.70	26.71	17.79	79.19	72.35	54.95	45.05	14.45	ND	ND
ZLC	56.26	8.77	17.49	8.10	66.02	69.60	81.78	18.22	13.27	ND	ND
ZLC	62.56	19.08	22.34	12.95	83.03	62.42	82.71	17.29	14.66	6.25	6.23
RT	31.11	32.05	34.72	1.69	77.11	75.38	80.70	19.30	13.48	ND	ND
RT	32.06	29.46	22.36	2.86	82.07	76.27	74.88	25.12	12.32	ND	ND
RT	38.12	17.89	29.52	5.60	93.15	89.02	76.77	23.23	10.92	ND	ND
SMT	89.07	17.75	21.23	7.39	85.91	71.34	39.62	60.38	12.79	ND	ND
SMT	88.37	24.27	31.41	8.98	81.65	65.31	42.66	57.34	13.36	ND	ND
SMT	94.48	18.89	23.32	7.44	84.50	80.30	41.94	58.06	12.54	ND	ND
NVV	88.12	41.76	24.68	1.14	86.25	42.24	33.48	66.52	12.80	ND	ND
NVV	89.75	38.90	23.40	5.40	79.64	44.93	43.58	56.42	13.39	ND	ND
NVV	87.62	40.76	21.87	2.95	90.18	48.25	47.67	52.33	12.35	ND	ND
SJBO	57.46	32.72	70.26	13.25	88.72	70.19	42.49	57.51	8.91	ND	ND

TPC = Total phenolic content. F = Flavonoid content. DPPH = Free radical scavenging, ABTS = Free radical scavenging, MCA = Metal chelating activity, FRAP = Ferric reducing antioxidant power, CMS = Cell membrane stabilization. IPTD = Inhibition of protein thermal denaturation, H, Haemolysis, GP = Gram positive *Staphylococcus aureus* (MIC). GN = Gram negative *Salmonella typhi* (MIC), MIC = Minimum inhibitory concentration, ND = Not detected. EY = Espita, Yucatan; MY = Mama, Yucatan; CNC = Centauros del Norte, Campeche; IKC = Ich-Ek, Campeche; ZLC = Zoh-Laguna, Campeche; BQR = Bacalar, Quintana Roo; FCPQR = Felipe Carrillo Puerto, Quintana Roo; TQR = Tihosuco, Quintana Roo; RT = Reforma, Tabasco; SMT = San Marcos, Tabasco; NVV = Ejido Nicolás Bravo, Veracruz; SJBO = San Juan Bautista, Oaxaca.

**Table 3 foods-12-01434-t003:** Pearson correlation coefficients between various properties of propolis extracts.

Properties	TPC	FC	DPPH	ABTS	MCA	FRAP	CMS	IPTD	H	GP	GN
TPC	1										
FC	−0.426	1									
DPPH	0.892	−0.359	1								
ABTS	0.818	−0.549	0.810	1							
MCA	−0.622	0.397	−0.605	−0.649	1						
FRAP	0.546	−0.578	0.589	0.480	−0.094	1					
CMS	0.037	−0.291	0.001	0.236	−0.497	−0.018	1				
IPTD	−0.180	−0.013	−0.373	−0.330	0.183	−0.052	0.030	1			
H	−0.037	0.291	−0.001	−0.236	0.497	0.018	−1.000	−0.030	1		
GP	−0.114	0.264	−0.099	−0.130	0.093	0.022	−0.017	−0.142	0.017	1	
GN	−0.114	0.128	−0.106	−0.076	0.054	−0.129	0.055	−0.049	−0.055	0.647	1

TPC = Total phenolic content (mg equivalents of gallic acid/mL extract). F = Flavonoids (mg equivalents of catechin/mL extract). DPPH = DPPH free radical (hydrophobic) scavenging, ABTS = ABTS free radical (hydrophilic) scavenging, MCA = Metal chelating activity, FRAP = Ferric reducing antioxidant power, CMS = Cell membrane stabilization, IPTD = Inhibition of protein thermal denaturation, H = Hemolysis. GP = Gram-positive Staphylococcus aureus (MIC). GN = Gram-negative Salmonella typhi (MIC), MIC = Minimum inhibitory concentration.

## Data Availability

Data is contained within the article.

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
