# Peer review of "Phenolic Content and Bioactivity as Geographical Classifiers of Propolis from Stingless Bees in Southeastern Mexico"

_foods, 2023, doi:10.3390/foods12071434_

Round 1

Reviewer 1 Report

The aim of this study was to classify Melipona beecheii propolis according to its phenolic compound  content and potential pharmacological activity, using in vitro assays and statistical methodologies of multivariate analysis, hierarchical cluster analysis and principal component analysis.

Please address the following concerns:

Line 86 – I advise the authors to rephrase the subtitle (i.e. Propolis samples collection).

Lines 104 and 108 – Please check the expression of concentrations “(70% w/v)” and “70% ethanol/water”.

Lines 386-411 – Please expand the Discussion section of your manuscript and provide an analysis of the results you obtained by comparing your results to the ones that were published in other studies. A correlation between your results should be included in this section as well.

Author Response

Reviewer 1

The authors of the manuscript appreciate the reviewer's comments.

Responses to reviewer comments:

Line 86 – I advise the authors to rephrase the subtitle (i.e., Propolis samples collection).

The subtitle has been changed.

Lines 104 and 108 – Please check the expression of concentrations “(70% w/v)” and “70% ethanol/water”.

Concentration expressions were modified.

Lines 386-411 – Please expand the Discussion section of your manuscript and provide an analysis of the results you obtained by comparing your results to the ones that were published in other studies. A correlation between your results should be included in this section as well.

The discussion section was improved considering the reviewer's comments.

* All changes made to the manuscript were highlighted for ease of review.

Reviewer 2 Report

Review report:

Title: Phenolic content and bioactivity as geographical classifiers of 2 propolis from stingless bees in southeastern Mexico

Reference. ID: foods- 2245154-

-The authors tried to classify Melipona beecheii propolis from 21 samples collected from seven states in southeastern Mexico according to their total phenolic content and potential pharmacological activity, using in vitro assays and statistical analysis.

The main idea is useful and interesting but the main - main critical remarks related to this work are :

-lack of phytochemical data of the samples (phenolic profiles and essential oil) this allow a better discussion of results. This the

-the statistical analysis should be restructured and improved.

-The beginning of the abstract seems as an introduction, Lines 14-18, the main idea should be reduced to 2 sentences.

-Line 19 and throughout the MS change “phenolic compound  content” as ’total phenolic content’, to avoid confusion with individual phenolic compounds

-line 27 change bioactivity as ‘in vitro bioactivity potential

-Line 85, HNO3’, HNO3

-line 86, un extra dot to delete : 2.2. .Propolis

-Fig 1 is not of good quality, is it hand drawn, it could be improved and added as supplementary data.

-line 110, it would be better to express TPC and TFC as mg/ g of propolis, why the authors chose to present in mg/mL extract ?

-lines 123, 133, 142, 154, 176, all equation should be edied using equation editor available in word

Lines 2.7.1, a reference should be added to the protocol.

2.8. indicate in this section which Yi were retained for ACP and HCA analysis.

-table 1 : code sample should be added in footnote. It is better to change table 1 by a moustache figure shouwing average value with min and max variation of each state (one figure Xa with 6 moustache) and Figure Xb with 12 moustaches for each locality.

-All localities abbreviation should be mentioned as footnote.

-3.2 antioxidant assays,

-Introductions sections to define the aim of each antioxidant assay in unnecessary. -Please avoid lines 233-234, lines 239-241, lines 246-248, lines 264-266, lines 270-272,lines  and linesand try to discuss results instead of explaining the aim of the assay which is obvious.

-Lies 255, ferric with capital letter F

-Line 303, title 3.5. statistical analysis is inappropriate, change as multivariate analysis or other title identical to 2. 8 line 199 2.8. Statistical analysis

Figure 2, add in footnote all localities abbreviation,

-Fig 3 and fig 4 same the same result, you can retain fig 3 and put observation in black and yi (variables) in red for example to differentiate samples from variables

-idea in lines 396-399 should be clarified: However, the fact that 397 some samples with low contents of phytochemicals exhibited high values of antioxidant 398 activity and vice versa, indicates that the composition of the extracts could influence more 399 than the content of extracts [48-49].??

-HCA and PCA (fig 2, fig 3-4 give the same idea results, clusters), if this is the case, why the author use both analysis, ? if not what is the specific extra information is given by each analysis.

-discussion should be more supported including proplis proximate composition if sample if available (at least: % balsam, % wax, % essential oils, and %  pollen

-Activity are not only correlated to phenolic content and profiles but also to quantity and quality of essential oil,

-If any phytochemical data are available of the sample (phenolic profiles and essential oil) this allow a better discussion of results. This the main critical remark related to this work besides of a better presentation of ACP and results

Author Response

Reviewer 2

The authors of the manuscript appreciate the reviewer's comments.

Responses to reviewer comments:

-Lack of phytochemical data of the samples (phenolic profiles and essential oil) this allow a better discussion of results.

The present study focused specifically on the total phenolic content and flavonoid content since several authors report them as the phytochemicals most related to in vitro bioactivities potential such as antioxidant, anti-inflammatory and antimicrobial activities. Additionally, these phytochemicals have been used together with other variables to establish the geographical origin and classification of other stingless bee products.

-The statistical analysis should be restructured and improved.

The statistical analysis of the variables was revised, and their description improved in the results and discussion sections.

-The beginning of the abstract seems like an introduction, Lines 14-18, the main idea should be reduced to 2 sentences.

The abstract was revised and improved.

-Line 19 and throughout the MS change “phenolic compound content” as ’total phenolic content’, to avoid confusion with individual phenolic compounds.

The modification suggested by the reviewer was made.

-Line 27 change bioactivities as ‘in vitro bioactivity potential

The modification suggested by the reviewer was made.

-Line 85, HNO3’, HNO3

The modification suggested by the reviewer was made.

-Line 86, un extra dot to delete: 2.2. Propolis

The modification suggested by the reviewer was made.

-Fig 1 is not of good quality, is it hand drawn, it could be improved and added as supplementary data.

The map was removed and replaced with a table describing collected states, locations, symbols, and geographical coordinates.

-Line 110, it would be better to express TPC and TFC as mg/ g of propolis, why the authors chose to present in mg/mL extract?

The modification suggested by the reviewer was made.

-Lines 123, 133, 142, 154, 176, all equation should be eddied using equation editor available in word.

The modification suggested by the reviewer was made.

-Lines 2.7.1, a reference should be added to the protocol.

A reference was added to the protocol.

-2.8. indicate in this section which Yi were retained for ACP and HCA analysis.

The strategies followed for the analysis of the variables are described in the results and discussion sections.

-Table 1: code sample should be added in footnote. It is better to change table 1 by a moustache figure shouwing average value with min and max variation of each state (one figure Xa with 6 moustache) and Figure Xb with 12 moustaches for each locality.

Sample codes have been added. The table is maintained to present the full data in accordance with reports by other authors who present data in this way in studies on classification, authenticity, and determination of geographical origin of propolis and stingless bee honey.

-All localities abbreviation should be mentioned as footnote.

The modification suggested by the reviewer was made.

-3.2 antioxidant assays,

-Introductions sections to define the aim of each antioxidant assay in unnecessary. -Please avoid lines 233-234, lines 239-241, lines 246-248, lines 264-266, lines 270-272, lines  and lines and try to discuss results instead of explaining the aim of the assay which is obvious.

The modification suggested by the reviewer was made.

-Lies 255, ferric with capital letter F

The modification suggested by the reviewer was made.

-Line 303, title 3.5. statistical analysis is inappropriate, change as multivariate analysis or other title identical to 2. 8 line 199 2.8. Statistical analysis.

The modification suggested by the reviewer was made.

-Figure 2, add in footnote all localities abbreviation.

The modification suggested by the reviewer was made.

-Fig 3 and fig 4 same the same result, you can retain fig 3 and put observation in black and yi (variables) in red for example to differentiate samples from variables.

The modification suggested by the reviewer was made.

-Idea in lines 396-399 should be clarified: However, the fact that 397 some samples with low contents of phytochemicals exhibited high values of antioxidant 398 activity and vice versa, indicates that the composition of the extracts could influence more 399 than the content of extracts [48-49].??

The modification suggested by the reviewer was made.

-HCA and PCA (fig 2, fig 3-4 give the same idea results, clusters), if this is the case, why the author uses both analysis? if not what is the specific extra information is given by each analysis.

The figures are complementary and support the results and discussion sections. In addition, both figures facilitate the reader's understanding of the results.

-Discussion should be more supported including propolis proximate composition if sample if available (at least: % balsam, % wax, % essential oils, and %pollen.

Composition results were not included as they will be presented in a specific article on physicochemical composition.

-Activity are not only correlated to phenolic content and profiles but also to quantity and quality of essential oil.

This study did not include the study of the correlation of bioactivities with essential oil content.

-If any phytochemical data are available of the sample (phenolic profiles and essential oil) this allow a better discussion of results. This the main critical remark related to this work besides of a better presentation of ACP and results.

Phenolic profiles and other compounds such as essential oils will be studied and reported in a future manuscript.

* All changes made to the manuscript were highlighted for ease of review.

Round 2

Reviewer 2 Report

The authors respond to the main comments.  The results of physicochemical composition of the samples, essential oil and main phenolic compounds are important to correlate with observed activities.

All equation should be numbered and written using equation editor, exponent, indices,..

Author Response

All equations were numbered and written using the equation editor.